# Role of Muscle Ultrasound for the Study of Frailty in Elderly Patients with Diabetes: A Pilot Study

**DOI:** 10.3390/biology12060884

**Published:** 2023-06-19

**Authors:** Andreu Simó-Servat, Ernesto Guevara, Verónica Perea, Núria Alonso, Carmen Quirós, Carlos Puig-Jové, María-José Barahona

**Affiliations:** 1Department of Endocrinology and Nutrition, Hospital Universitari Mútua Terrassa, 08221 Terrassa, Spain; 2Department of Medicine, Autonomous University of Barcelona, 08193 Barcelona, Spain; 3Department of Geriatrics, Hospital Universitari Mútua Terrassa, 08221 Terrassa, Spain

**Keywords:** musculoskeletal ultrasound, diabetes, sarcopenia, frailty, body composition, bioelectrical impedance analysis

## Abstract

**Simple Summary:**

It is well known that diabetes and sarcopenia are risk factors for developing frailty. The latest consensus on the management of older people with diabetes recommends routine frailty screening in check-up consultations for these patients. Therefore, there is a need to find more precise imaging techniques to determine sarcopenia and to distinguish site-specific sarcopenia—in this particular case, the quadriceps—which allows deambulation. Ultrasound technology is a low-cost, fast, and accessible technique in consultations to study body composition in patients with frailty. In this study, our aim was to validate muscle ultrasounds to complement physical fragility diagnoses using specific tests, such as the SARC-F, bioelectrical impedance analysis (BIA), and dynamometer. In fact, we demonstrated that muscle ultrasounds correlated with the BIA and could, therefore, be a useful tool for identifying regional sarcopenia of the quadriceps in elderly patients with diabetes. As this was a pilot study, larger studies to validate this simple and affordable screening method are necessary.

**Abstract:**

Background: Sarcopenia and diabetes contribute to the development of frailty. Therefore, accessible methods, such as muscle ultrasounds (MUSs), to screen for sarcopenia should be implemented in clinical practice. Methods: We conducted a cross-sectional pilot study including 47 patients with diabetes (mean age: 77.72 ± 5.08 years, mean weight: 75.8 kg ± 15.89 kg, and body mass index: 31.19 ± 6.65 kg/m^2^) categorized as frail by the FRAIL Scale or Clinical Frailty Scale and confirmed by Fried’s Frailty Phenotype or Rockwood’s 36-item Frailty Index. We used the SARC-F questionnaire to identify sarcopenia. The Short Physical Performance Battery (SPPB) and the Timed Up and Go (TUG) tests were used to assess physical performance and the risk of falls, respectively. In addition, other variables were measured: fat-free mass (FFM) and Sarcopenia Risk Index (SRI) with the bioimpedance analysis (BIA); thigh muscle thickness (TMT) of the quadriceps with MUS; and hand-grip strength with dynamometry. Results: We observed correlations between the SARC-F and FFM (R = −0.4; *p* < 0.002) and hand-grip strength (R = −0.5; *p* < 0.0002), as well as between the TMT and FFM of the right leg (R = 0.4; *p* < 0.02) and the SRI (R = 0.6; *p* < 0.0001). We could predict sarcopenia using a logistic regression model with a ROC curve (AUC = 0.78) including FFM, handgrip strength, and TMT. The optimal cut-off point for maximum efficiency was 1.58 cm for TMT (sensitivity = 71.4% and specificity = 51.5%). However, we did not observe differences in the TMT among groups of greater/less frailty based on the SARC-F, SPPB, and TUG (*p* > 0.05). Conclusions: MUSs, which correlated with the BIA (R = 0.4; *p* < 0.02), complemented the diagnosis, identifying regional sarcopenia of the quadriceps in frail patients with diabetes and improving the ROC curve to AUC = 0.78. In addition, a TMT cut-off point for the diagnosis of sarcopenia of 1.58 cm was obtained. Larger studies to validate the MUS technique as a screening strategy are warranted.

## 1. Introduction

Frailty is defined as age-related deterioration in physiological reserves across various organ systems, which increases the risk for injuries and fractures due to falls, acute illness, hospitalization, and mortality [1,2,3,4,5]. An important phenotype of frailty is sarcopenia, one of the most important geriatric syndromes. Sarcopenia is described by the European Working Group on Sarcopenia in Older People 2 (EWGSOP2) [6] as the loss of muscle function and muscle mass, which can be worse in patients with diabetes and obesity. A new meta-analysis [7] showed the presence of sarcopenia to be associated with an increased risk (odds ratio [OR]: 1.38 [95% confidence interval [95%CI]: 1.27–1.50]) of type 2 diabetes (T2DM). The exact underlying mechanism is still not fully understood, but there seems to be a bidirectional relationship involved, with chronic inflammation and insulin resistance emerging as key factors [8]. Therefore, assessing sarcopenia in the aging diabetic population is important for predicting the risk of morbidity and mortality [9]. The primary objective for older patients is to enhance their quality of life, making early detection of frailty and sarcopenia crucial aspects. Moreover, there is a high prevalence of sarcopenia among patients with diabetes, ranging from 7% to 29.3% [10]. The considerable variation in sarcopenia prevalence can be attributed to various factors, such as the diagnostic criteria employed and the diverse methods used for quantitative evaluation [11].

Ultrasounds (USs) are emerging as a promising alternative tool to assess sarcopenia through the measurement of thigh muscle thickness (TMT) due to their safety, noninvasiveness, low cost, and real-time characteristics [12]. Computed tomography (CT), magnetic resonance (MR), and dual-energy X-ray absorptiometry (DEXA) are not applicable in everyday clinical practice since they are expensive and often uncomfortable for patients [13]. The bioimpedance analysis (BIA) is a preferred alternative but implies a certain degree of inaccuracy in clinical practice because it depends on hydration status. Indeed, in the most recent systematic review aimed at defining sarcopenic obesity, 21 out of 75 studies employed the BIA for this specific objective [14]. In this scenario, innovative tools such as skeletal muscle ultrasounds (MUSs) for screening and diagnosing sarcopenia could be standardized, as they are proved to be an accurate imaging technique to implement into clinical practice for diagnosis and follow-up in the specific elderly population with diabetes. Furthermore, in the context of endocrinology departments, the use of US technology offers clear advantages since the same probe employed for thyroid USs can be utilized for screening for sarcopenia. Additionally, the BIA and dynamometers are readily accessible in clinics. Therefore, employing these three techniques enables a more accurate characterization of patients, offering significant morphological details regarding important muscle groups, such as the quadriceps [15]. These methods can provide valuable information concerning functionality. For instance, a previous study demonstrated a link between the thickness of the quadricep femoris and the isometric maximum voluntary contraction force [16]. In fact, the majority of previous investigations of MUSs in the lower extremities have primarily focused on exploring the relationship between muscle strength and muscle thickness [17]. Minetto et al. [18] proposed that an MUS could serve as a practical tool for identifying individuals with low muscle mass. They examined the association between the thickness of the rectus femoris muscle using a US and the appendicular muscle mass measured using whole-body DEXA. However, the cut-off points of MUS parameters for diagnosing sarcopenia have not been established [19]. Thus, in the present paper, we aim to investigate the validity and the role of MUSs of quadriceps for their importance in deambulation, establishing a cut-off for TMT to predict sarcopenia and exploring how they complement the BIA and dynamometer analyses with the ROC curve. To better define the results obtained, we specifically conduct the study on an elderly population with diabetes. Moreover, we try to show the associations between MUSs and frailty physical tests.

## 2. Materials and Methods

This was a pilot cross-sectional study including 47 elderly patients with diabetes carried out in our hospital. Subjects were recruited in endocrinology consultations between March 2022 and January 2023. The study followed the STROBE guidelines for cross-sectional studies [20]. The exclusion criteria included individuals below the age of 70; patients with certain characteristics (such as drug or alcohol addiction and severe psychological or psychiatric disorders) that implied difficulty in monitoring; and patients with a history of traumatic injury, spinal injury, or any condition that could impact limb motor function. Patients were classified as frail according to screening scales (FRAIL Scale or Clinical Frailty Scale) and confirmed with a diagnosis of physical frailty (assessed with Fried’s Frailty Phenotype) or frailty due to accumulation of deficits (assessed with Rockwood’s 36-item Frailty Index). In this way, the inclusion criteria were age over 70 years, diabetes, and frailty demonstrated in geriatric tests. The hospital ethics committee approved all the procedures carried out in the study, and all subjects signed the informed consent.

### 2.1. Evaluation of Sarcopenia

Sarcopenia diagnostic methods can be differentiated into qualitative or quantitative since it is defined as loss of mass and muscle function. Following the EWGSOP2 consensus (Figure 1), the first step for the diagnosis of sarcopenia risk was the use of the SARC-F questionnaire, where a score of four or higher indicated probable sarcopenia [8]. The questionnaire consisted of five questions to determine the ability to perform routine activities (Figure 2a). Secondly, we used hand-grip strength with a dynamometer to evaluate muscle strength. To evaluate body composition, we measured total body fat-free mass (FFM) and right leg FFM with the BIA, as well as the TMT of the right quadriceps with an MUS. We also used the Sarcopenia Risk Index (SRI) calculated with the BIA, which measured the appendicular muscle mass to assess the risk of sarcopenia in people over 65 years. Subsequently, the Short Physical Performance Battery (SPPB) (physical performance test) and the Timed Up and Go (TUG) (risk of falls test) (Figure 2b,c) tests were carried out to evaluate physical state.

### 2.2. BIA

The FFM and SRI (range: 0 to 20; more risk when closer to 0) were measured using the BIA. A standing-position 8-electrode MF-BIA segmental body composition analyzer (MC-780MA; TANITA, Tokyo, Japan) was used, as in previous literature [25].

### 2.3. Dynamometer

Strength was measured using a Jamar Plus Digital Hand Dynamometer (Performance Health, Nottinghamshire, UK) while the patient was sitting in the anatomical position in a chair with back support and fixed arm rests with forearm flexed 90° without touching the body. The patient was asked to squeeze the Jamar as long and as tightly as possible, alternating between the right and left hands. We recorded three measurements for each hand, with a 10–20 s pause between each repetition to avoid the effects of muscle fatigue. The result was the highest value in kilograms.

### 2.4. Ultrasonic Technique

Since sarcopenia mainly affects the lower extremities, we used a US with the patient lying on a stretcher, relaxed, and looking up in accordance with the previous literature [26,27,28,29]. We evaluated TMT, performing minimal pressure to avoid excessive compression of the muscle and with the appropriate contact gel, adding the distance of the rectus femoris and vastus intermedius [30] (Figure 3). US measurements, according to the recommendations of the European Union Geriatric Medicine Society Sarcopenia Special Interest Group [28], were made with a sonographic US Logiq P9 (GE Healthcare) muscle-skeleton B-model instrument using a linear multifrequency transducer (4–11 Hz). The average value of a set of three consecutive measurements was calculated to assess the TMT. The data were reported in centimeters (cm) as mean values +/− standard deviation. The same physician (endocrinologist A.S.-S.) performed all measurements, as previously described [31].

### 2.5. Statistical Analysis

Categorical variables were defined with percentages and continuous variables as means ± standard deviation (SD) unless specified otherwise. On one hand, to assess the differences between groups we used a T-test for continuous variables. On the other hand, Pearson’s correlation test was employed to investigate the relationships between variables. The areas under the curve (AUCs) in the receiver operating characteristic (ROC) curve were calculated to examine the relationships among FFM, hand-grip strength, TMT, and sarcopenia as defined by the SARC-F. We utilized the Youden index to determine the cutoff value for identifying sarcopenia using TMT. The Youden index assigned equal importance to false positive and false negative results. All analyses were conducted with a two-tailed approach, considering p < 0.05 as the threshold for statistical significance [31]. We used STATA statistical software version 14 (College Station, TX, USA).

## 3. Results

A total of 47 subjects with insulin-treated diabetes were included, with 32 female participants (68%) and a mean age of 77.72 ± 5.08 years. The duration of diabetes was 28.81 ± 13.67 years, with a mean HbA1c of 8.22 ± 1.81%. The mean BMI was 31.19 ± 6.65 kg/m^2^. Measurements observed using the BIA, MUSs, and dynamometer are shown in Table 1.

To support and add value to the screening of sarcopenia with the SARC-F questionnaire, a significant inverse correlation was observed between muscle measurement values using FFM and test punctuations (R = −0.5, *p* < 0.002) and between hand-grip strength and test punctuations (R = −0.5, *p* < 0.0002). On the other hand, to validate MUS technology as a reliable method, significant correlations were observed between the TMT of the right quadriceps and the FFM of the right leg (R = 0.4, *p* < 0.02) and the SRI (R = −0.6, *p* < 0.0001), both calculated using the BIA. (Figure 4a–d).

However, we did not find statistically significant differences (*p* > 0.05) when analyzing TMT among patients considered sarcopenic using the SARC-F, SPPB, and TUG questionnaires; rather, we found a trend toward significance, correlation, and the mean (Table 2).

To examine the capacity of each method to predict sarcopenia using the SARC-F, we performed a ROC analysis, and the AUCs were as follows: for BIA, it was 0.64 (95% CI: 0.47845 to 0.80266), while it was 0.74 (95% CI: 0.58628 to 0.89857) for the dynamometer and 0.56 (95% CI: 0.40832 to 0.70498) for the MUS. Then, we performed a logistic regression model with the three methods, and we assessed the model performance with a ROC curve (AUC = 0.78) (Figure 5a–d). According to the ROC results and using the Youden index, the optimal cut-off point for maximum efficiency was 1.58 cm in TMT, as assessed with MUS (sensitivity = 71.4% and specificity = 51.5%; positive predictive value =38.5% and negative predictive value = 81%). Likewise, the patients of the study with a TMT less than 0.98 cm had a 100% probability of sarcopenia (sensitivity = 100% and specificity = 6.06%).

## 4. Discussion

There is growing awareness of the significance of sarcopenia in individuals with T2DM, both from clinical and research perspectives. The reason is the impact that sarcopenia can have on the quality of life of elderly individuals with T2DM, where frailty and sarcopenia are emerging as a category of complications [32]. Moreover, diabetes is closely linked to losses in muscle mass, strength, and functionality, which ultimately contribute to the development of sarcopenia, frailty, and disability. The current study provides evidence that MUSs are significantly related to the BIA, a standardized method to analyze body composition in patients with diabetes and sarcopenia or in patients at risk for it [33]. We found a significant correlation between the FFM of the right leg measured using the BIA and the TMT of the right quadriceps measured with an MUS (R = 0.4, *p* < 0.02). In addition, we observed a significant inverse correlation between MUSs and the SRI (lower results meaning more risk) measured with the BIA (R = −0.6, *p* < 0.0001).

On the other hand, we demonstrated inverse correlations between the SARC-F questionnaire (a score lower than four meaning healthy) and total FFM (R = −0.5, *p* < 0.002) and hand-grip strength (R = −0.5, *p* < 0.0002). These results support the EWGSOP2 consensus document that recommends these methods for identifying and diagnosing sarcopenia [8,21], as we used them in our sample. In this scenario, we established a cut-off value of <1.58 cm for TMT assessed with MUS to diagnose sarcopenia in our population (sensitivity = 71.4% and specificity = 51.5%). Moreover, we observed that a TMT less than 0.98 cm was able to predict sarcopenia in 100% of the cases, thus suggesting that MUS technology is a reliable method for screening for regional sarcopenia in the elderly population with diabetes. It is a promising result for a screening test but, of course, further studies are required, not only to compare using the BIA. Nevertheless, methods such as DEXA, CT, and MR are challenging to access and are expensive, not to mention the known radiation risks associated with some of them. Additionally, in many healthcare centers, body composition studies are not routinely performed or standardized, likely due to resource limitations. In contrast, MUS technology is an increasingly accessible tool in clinical settings, which can serve as a screening method for sarcopenia and provide valuable information on body composition. MUS technology is rapid, cost-effective, noninvasive, and safe, making it a favorable alternative [34]. Specifically, MUS technology can provide us with regional information about a muscle group, and it can be a complementary technique to both the BIA and the dynamometer. Therefore, our results point to further exploration of the usefulness of US technology and establishing cut-off points as a diagnostic method for regional sarcopenia, as detailed above. Furthermore, based on a regression logistic model with the three methods, we found better results in the ROC analysis (AUC = 0.78) than exclusively with the BIA (AUC = 0.64) or dynamometry (AUC = 0.74), demonstrating better screening using MUS technology. These findings are probably better due to quadriceps’ site-specific sarcopenia measurements, which could be missed with the other tests. It is known that the BIA can be affected by factors such as BMI or hypervolemia [13]; in fact, our sample was composed of a majority of patients with obesity (BMI = 31.19 ± 6.65 kg/m^2^). Furthermore, the BIA cannot be used in patients with pacemakers due to the risk of deprogramming, which does not happen when using US technology. Regarding dynamometry, it should be noted that hand-grip strength measured using dynamometry is clearly not related to the strength of the lower extremities.

In addition, to open new fields of research, US technology allows us to analyze muscle quality through echogenicity. We suggest using MUS assessment of lower-limb muscle thickness as an auxiliary criterion for evaluating dynapenia (muscle strength loss despite having normal skeletal muscle volume) [35]. In fact, a patient could have BIA results without sarcopenia but may present regional sarcopenia or dynapenia in the quadriceps, which is a very important muscle for a person’s autonomy and mobility. It is known that quadriceps’ thickness plus echogenicity provides a useful surrogate of force [36,37].

The main limitation of our study was the small sample size, as it was a pilot study. This is probably the reason why we did not reach a significant correlation or significant differences among groups of frailty in the physical frailty tests performed (the SARC-F, SPPB, and TUG) and the TMT. Although the results tended toward significance and correlation, we feel compelled to perform larger studies to validate MUS usage. Moreover, the drawbacks of US technology primarily include the absence of standardized protocols and its heavy reliance on the proficiency and capabilities of the operator [38]. Secondly, the interpretation of muscle–fat interfaces was restricted because these tissues have similar acoustic impedance, making it challenging to differentiate between them accurately using US technology. Another limitation of the MUS is the potential for measurement errors caused by the operator applying excessive pressure with the transducer on the skin, which can compress the muscle and lead to inaccuracies [25,39]. Thirdly, the lack of different physicians conducting the measurements prevented test reproducibility. We only took the TMT of the anterior compartment of the quadriceps; muscle area or pennation angle are probably important but may implicate time-consuming consultations, and our objective was to achieve efficiency in clinical practice. On the other hand, the echogenicity of muscle needs a software not implemented and validated yet for diagnosis. Finally, larger studies are needed to confirm our results, mainly because a significant error regarding gold standard measurements could be present.

To conclude, the role of MUS measurements of TMT to identify regional sarcopenia in elderly patients with diabetes was reliable and complemented both the methods established in clinical practice and frailty tests.

## 5. Conclusions

A major cause of frailty is sarcopenia [40], and its prevalence is 25.6% in people with T2DM [33]. From another point of view, diabetes is associated with an accelerated aging process that promotes frailty, and this is linked to an increased risk of sarcopenia [40,41]. In this scenario, it is imperative to increase the ability of physicians to recognize the presence of sarcopenia in T2DM patients, particularly in older patients. In fact, in people with diabetes, sarcopenia is emerging as a third category of complications in addition to the traditional micro- and macrovascular diseases, leading to considerable disability [42]. The TMT of the muscle can be utilized to confirm the existence of muscle mass depletion and exhibits correlations with DEXA, CT, and MR measurements [43]. As discussed, the identification of site-specific muscle loss through US, known as “regional” or “site-specific” sarcopenia, should be incorporated in sarcopenia assessment. Given that sarcopenia can manifest in a site-specific manner with greater muscle loss observed in the lower limbs compared to the upper limbs, evaluating the anterior compartment of the thigh can be considered an appropriate anatomical region for muscle ultrasounds. Therefore, this study allowed progress in the investigation to establish the validity of muscle ultrasounds for this field.

In this study, we demonstrated that muscle ultrasounds correlated with bioelectrical impedance and complemented it as a useful tool for identifying regional sarcopenia of the quadriceps in elderly patients with diabetes. Moreover, it probably improved the diagnosis of physical frailty through specific tests and questionnaires. However, it is still a pilot study and, therefore, larger studies to validate this simple, low-cost screening strategy are needed to confirm our results.

## Figures and Tables

**Figure 1 biology-12-00884-f001:**
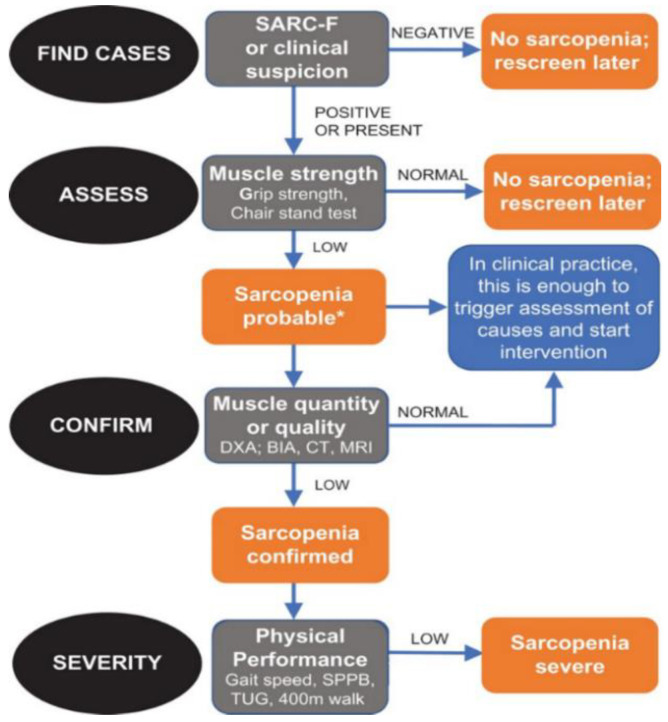
Sarcopenia: EWGSOP2 algorithm. Modified from Cruz-Jentoft A.J. et al. [21]. * DXA: dual-energy X-ray absorptiometry; BIA: bioimpedance analysis; CT: computed tomography; MRI: resonance; SSPB: Short Portable Performance Battery; TUG: Timed Up and Go.

**Figure 2 biology-12-00884-f002:**
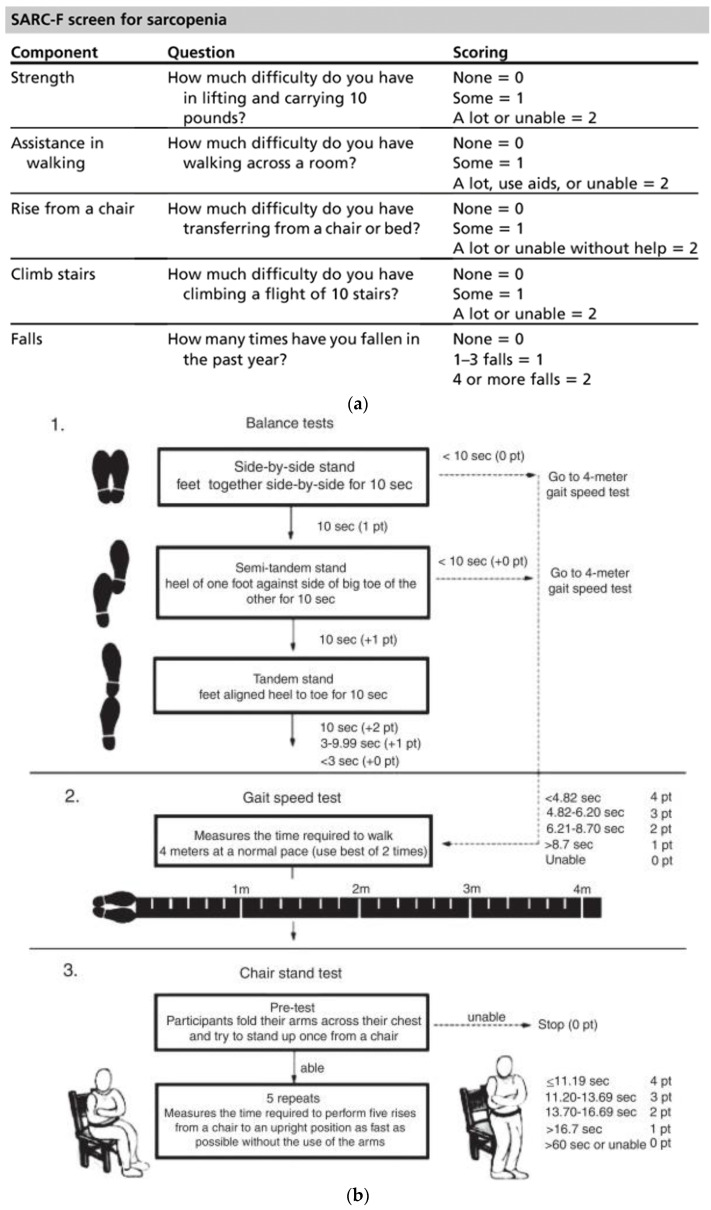
(**a**) SARC-F questionnaire [22]; (**b**) SPPB test [23]; (**c**) Timed Up and Go [24].

**Figure 3 biology-12-00884-f003:**
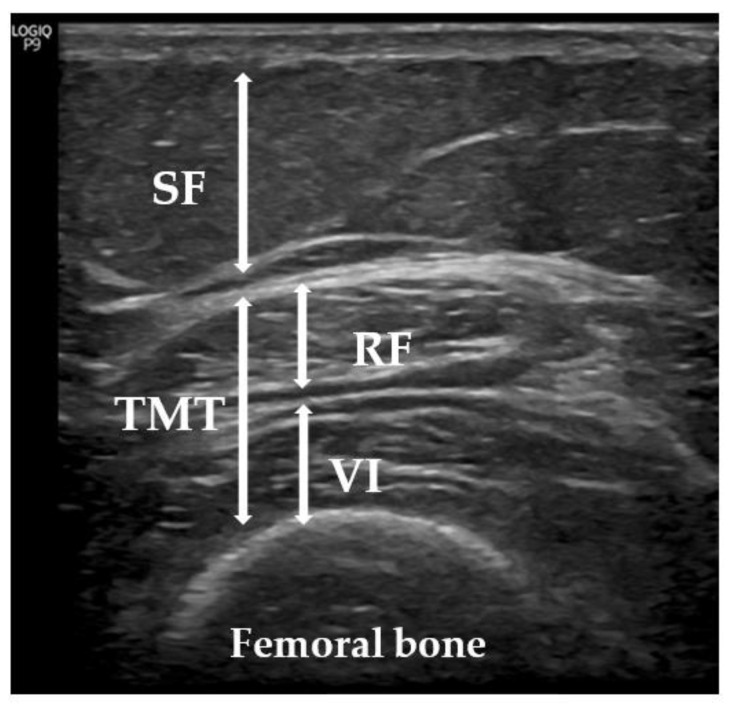
Measurement of subcutaneous tissue and thigh muscles using US. Representative image of the quadriceps from a participant in our sample. SF: subcutaneous fat; TMT: thigh muscle thickness; VI: vastus intermedius; RF: rectus femoris.

**Figure 4 biology-12-00884-f004:**
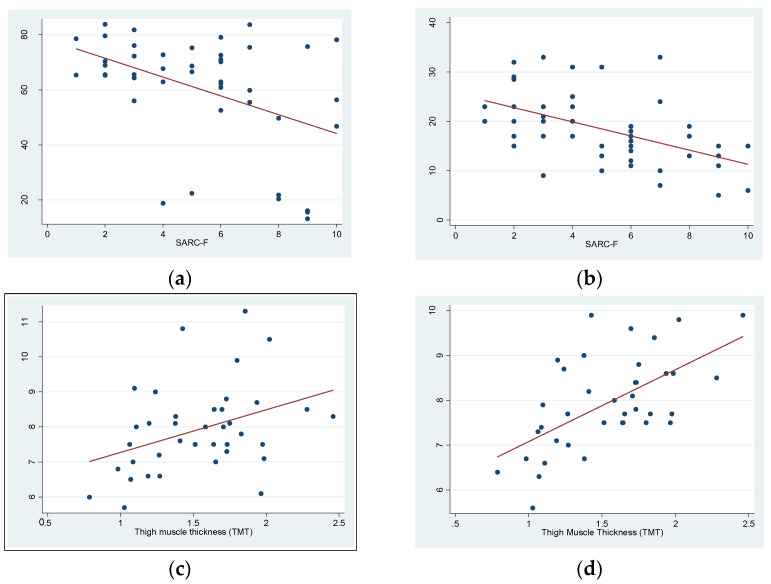
(**a**) A significant inverse correlation was observed between fat-free mass (FFM) from the bioimpedance analysis (BIA) and the SARC-F questionnaire. (**b**) A significant inverse correlation was observed between dynamometer hand-grip strength and the SARC-F questionnaire. (**c**) A significant direct correlation was observed between fat-free mass (FFM) of the right leg and thigh muscle thickness (TMT) assessed by ultrasound. (**d**) A significant inverse correlation was observed between the Sarcopenia Risk Index (SRI) and thigh muscle thickness (TMT) assessed by ultrasound.

**Figure 5 biology-12-00884-f005:**
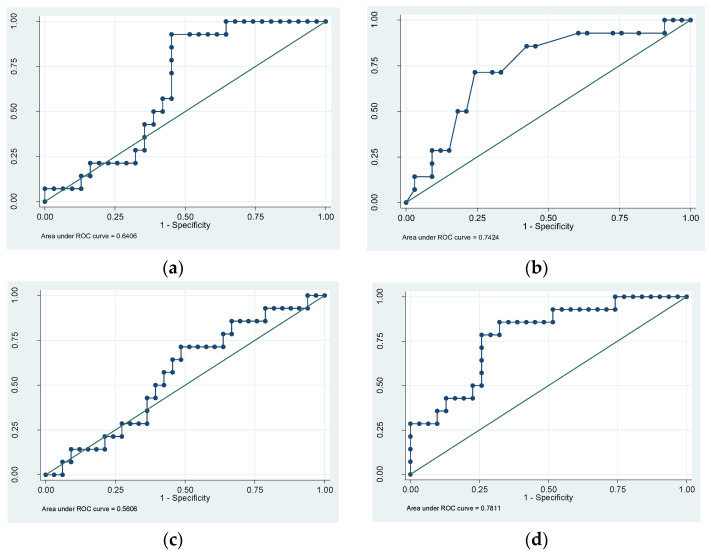
(**a**) To assess the capacity of the bioimpedance analysis (BIA) to predict sarcopenia using the SARC-F questionnaire, we performed a ROC analysis (AUC = 0.64). (**b**) To assess the capacity of the dynamometer to predict sarcopenia using the SARC-F questionnaire, we performed a ROC analysis (AUC = 0.74). (**c**) To assess the capacity of muscle ultrasounds (MUSs) to predict sarcopenia using the SARC-F questionnaire, we performed a ROC analysis (AUC= 0.56). (**d**) Performance of the logistic regression model with the three methods using the ROC curve (AUC = 0.78).

**Table 1 biology-12-00884-t001:** Measurements with BIA, MUSs, and dynamometer.

	Mean ± SD
FFM * total body (%)	60.41 ± 20.19
FFM right leg (%)	7.92 ± 1.25
SRI *	7.94 ± 1.04
TMT right quadriceps (cm)	1.55 ± 0.4
Dynamometer (kg) **	14.84 ± 5.02 (female) and 25.03 ± 6.42 (male)

* FFM: fat-free mass; SRI: Sarcopenia Risk Index; TMT: thigh muscle thickness. ** Considered sarcopenic by dynamometer: 16 kg (female) and 27 kg (male).

**Table 2 biology-12-00884-t002:** Comparison of means among frailty groups in the three tests carried out.

	SARC-F < 4	SARC-F > 3 **	SPPB * < 7 **	SPPB > 6	TUG < 20	TUG * > 19 **
n	15	32	33	14	20	18
Mean ± SD (cm)	1.59 ± 0.36	1.54 ± 0.42	1.49 ± 0.40	1.57 ± 0.40	1.54 ± 0.39	1.45 ± 0.23
p	0.2		0.09		0.6	
R	0.2		−0.25		0.1	

* SPPB: Short Physical Performance Battery; TUG: Timed Up and Go. ** Frailty group: >3 in SARC-F, <7 in SPPB, and >19 in TUG.

## Data Availability

Not applicable.

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
