# Peer review of "Role of Muscle Ultrasound for the Study of Frailty in Elderly Patients with Diabetes: A Pilot Study"

_biology, 2023, doi:10.3390/biology12060884_

Round 1
Reviewer 1 Report
Dear authors, thank you for submitting this excellent research manuscript. It is well conducted and well written. I believe it possesses a few weaknesses that ought to be addressed.
First, there are minor spelling mistakes that should be corrected.
In the introduction, the US and bioimpedance analysis should be discussed more thoroughly. How well can BIA truly diagnose sarcopenia, wouldn't comparison with DXA be more precise? Why is the US emerging as a promising tool for sarcopenia evaluation? What is known about US muscle measurements and sarcopenia?
I suggest authors to discuss why recognizing sarcopenia is important in DM patients?
The authors employed muscle US on rectus femoris for the evaluation of sarcopenia; however, solely subcutaneous fat and thigh muscle thickness were measured. There are several parameters that could also be evaluated as: cross-section, echogenicity, or penetration angle.. I would suggest authors to discuss this limitation of their study.
I suggest authors not to use abbreviations (MUS, BIA) in "conclusions"
First, there are minor spelling mistakes that should be corrected.
Reviewer 2 Report
Dear Authors,
Manuscript ID: biology- 2354042
Title Manuscript: Role of muscle ultrasound for the study of frailty in elderly patients with diabetes: a pilot study
This pilot cross-sectional study examined the muscle ultrasound for screening sarcopenia and body composition parameters in 47 elderly patients with diabetes. This pilot study is an interesting and important topic since the study participants are elderly patients with diabetes > 70 yrs but at the moment MAJOR REVISIONS are necessary:
POINTs of STRENGTH:
1) Assessment of sarcopenia index with muscle ultrasound in elderly patients with diabetes > 70 yrs.
POINTs of WEAKNESS (and/or should be revised to improve the manuscript):
Abstract
2) Please add mean age, weight and BMI for participants in the “methods” section of the abstract;
3) Please report the “conclusion section” based on the results obtained from the study;
Keywords
4) Please add words of “Body composition” and “Bioelectrical impedance analysis” in the keywords section.
1. Introduction
5) The hypothesis and purpose of this study can be stated in more detail, especially for body composition parameters and OR type of participants (elderly patients with diabetes > 70 yrs), and so on;
2. Materials and Methods
6) The recruitment process and/or screening of study participants, especially inclusion and exclusion criteria should be described in more detail such as initial sample size OR reasons for the withdrawal of participants from the study, gender, age, BMI, blood pressure, free of medications and/ or drug intervention, physical fitness level, VO2max, METs, healthy status/ metabolic disorders, and so on.
2.1. Evaluation of sarcopenia
7) Considering that the gold standard for evaluating body composition parameters and sarcopenia index is the DXA device, why did the authors not use the DXA device?
2.3. Dynamometer
8) Please specify model and company/country of dynamometer;
2.4. Ultrasonic Technique
9) Which technique measures more accurate results for sarcopenia? DXA OR Ultrasonic technique? Please explain;
2.4. Statistical analysis
10) Did authors use a statistical software to calculate the sample size? If YES, please explain and add its name and valid reference in the statistical analysis section.
3. Results
11) Please specify the Y-axis parameter (a-d) in Figure 4;
4. Discussion and Conclusion
12) As mentioned above, the authors will agree that the limitations section has to be expanded;
13) What does this study add to the literature? Please explain and add in the conclusions section;
14) Please provide clinical implications for elderly patients with diabetes OR other elderly patients with metabolic disorders at the end of this manuscript.
References
15) References are not always in accordance with the authors' guidelines. In particular, please check No. x and xv for validation.
Best Regards
21 April 2023
Round 2
Reviewer 1 Report
The authors satisfactorily addressed all my comments.
Reviewer 2 Report
Dear Authors,
Manuscript ID: biology- 2354042
Title Manuscript: Role of muscle ultrasound for the study of frailty in elderly patients with diabetes: a pilot study
I am very grateful to the authors for their efforts.
In general, this manuscript has found suitable content after correcting major revisions, and the modified revisions are accepted.
Best Regards
9 June 2023